# Chitosan-Based Materials: An Overview of Potential Applications in Food Packaging

**DOI:** 10.3390/foods11101490

**Published:** 2022-05-20

**Authors:** Tong Liu, Junbo Li, Qilong Tang, Peng Qiu, Dongxia Gou, Jun Zhao

**Affiliations:** College of Food Science and Engineering, Changchun University, Changchun 130022, China; liut@ccu.edu.cn (T.L.); lijunbo0111@163.com (J.L.); 200301046@mails.ccu.edu.cn (Q.T.); a17835093337@163.com (P.Q.); goudx@ccu.edu.cn (D.G.)

**Keywords:** chitosan, biological activity, modification, packaging, preservation

## Abstract

Chitosan is a multifunctional biopolymer that is widely used in the food and medical fields because of its good antibacterial, antioxidant, and enzyme inhibiting activity and its degradability. The biological activity of chitosan as a new food preservation material has gradually become a hot research topic. This paper reviews recent research on the bioactive mechanism of chitosan and introduces strategies for modifying and applying chitosan for food preservation and different preservation techniques to explore the potential application value of active chitosan-based food packaging. Finally, issues and perspectives on the role of chitosan in enhancing the freshness of food products are presented to provide a theoretical basis and scientific reference for subsequent research.

## 1. Introduction

Chitosan, with the scientific composition of (1,4)-2-amino-2-deoxy-D-glucose, is the only natural cationic alkaline polysaccharide. Chitin is mainly found in shrimp, crab [1], and insect shells [2]. Chitosan can be produced by the deacetylation of chitin with NaOH, as shown in Figure 1, and by fermentation with some microbial cultures [3]. As a renewable green bio-resource from a wide range of sources, chitosan has superior bacterial inhibition and antioxidant properties and enzyme inhibition activity, and it is edible, nontoxic, and biodegradable [4], so there is a great deal of research being carried out on it in the food, medicine, environmental protection, chemical, cosmetics, and textile fields, among others [5]. It has good prospects in food preservation and green packaging because of its antibacterial [6,7,8,9], antioxidant [10,11,12,13,14], and enzyme inhibiting [15,16,17,18] activities and biodegradability [19,20,21]. However, the tensile strength, structural strength, and physical properties, such as oxygen and water vapor transmission, after being made into film cannot fully meet the requirements of food packaging, so the performance of chitosan as a food packaging material must be improved by employing derivatization, preparing nanocomposites, and compounding it with biological extracts. Many studies have shown that the future direction of food preservation is to make it green, safe, and environmentally friendly preservation [22,23,24,25,26]. Therefore, the use of chitosan as a base material for food preservation has become a research hotspot [27,28]. Even so, research on chitosan to date has not actually resulted in a commercially available packaging product. This paper reviews reports on the relevant biological activities of chitosan in food preservation and discusses the potential applications of chitosan derivatization, chitosan-based nanocomposites, and chitosan–extract composites, as shown in Figure 2, to provide a reference for the application of chitosan in food preservation and the development of novel preservation technologies.

## 2. Research Methods

In this review, information on the biological activity of chitosan and food packaging based on chitosan was consulted through various literature databases and search engines on the Internet, including but not limited to X-MOL, Google Scholar, PubMed, CNKI, AbleSci, Scopus. Through the integration of the obtained information, the hypothesis of the biological activity mechanism of chitosan in the field of food preservation is proposed, and the advantages and disadvantages of chitosan in the preservation of fresh products by coating or making packaging films are evaluated.

## 3. Biological Activity of Chitosan

### 3.1. Bacteriostatic Properties

The broad-spectrum bacterial inhibitory properties of chitosan have been confirmed by in vitro experiments and complex food antimicrobial assays [29,30,31,32]. In the process of food preservation, the use of chitosan coating or film covering the surface of food can, to a certain extent, resist microbial attack, slow down food spoilage, and thus extend the shelf life of food. For example, Wei et al. [33] studied the effects of different concentrations of chitosan on the color and surface microbial populations of frozen duck skin. The results showed that chitosan could form oxygenation competition with color-presenting substances, and 3% chitosan could effectively reduce the total number of surface colonies of duck skin during storage. In addition to the effect of chitosan concentration on the inhibition effect, molecular weight, degree of deacetylation, and bacterial species have also been shown to have an effect. Benhabiles et al. [6] studied the effect of molecular weight and degree of deacetylation on the inhibition effect, comparing the inhibition performance of chitosan on four Gram-positive and seven Gram-negative bacteria. The results showed that chitosan with low molecular weight and a high degree of deacetylation had a better inhibition effect, and among the tested species, chitosan had an inhibition effect on all bacteria except *Salmonella typhimurium*. The pH of the solution and the presence of metal ions also have an effect on the bacterial inhibitory effect of chitosan. Jing et al. [34] used ANOVA to test the data and concluded that the inhibitory activity of chitosan increased with decreasing pH in the range of >4 and <6. Comparing the effects of metal ions on the inhibitory activity of chitosan at a pH of 4, the results showed that Ca^2+^ and Mg^2+^ significantly reduced the inhibitory ability against Gram-negative bacteria. It can be seen that the inhibitory activity of chitosan is related to a variety of factors, with interlocking effects among them, and the best inhibition results can be obtained only by comprehensively analyzing all of the factors.

Although the information on the antimicrobial activity of chitosan is controversial, it is generally accepted that yeasts and mycobacteria are the most sensitive organisms to chitosan, followed by Gram-positive and Gram-negative bacteria [35]. The antifungal activity of chitosan is mainly attributed to the inhibition of mycelial growth and spore germination [36]. Early studies showed that chitosan inhibits spore germination, budding tube elongation, and radial growth by sequestering metals, minerals, trace elements, or nutrients required for fungal growth [37]. The inhibition of pathogenic fungi increases with increased chitosan concentration [38], but its application is hindered by low solubility at physiological pH due to its low charge density. Therefore, increasing the positive charge density of chitosan may be the most effective way to improve its solubility and antifungal activity [39]. The antifungal activity of chitosan also depends on its molecular weight [40], which affects the morphogenesis of the fungal cell wall [41]. It has also been shown that chitosan’s inhibition of pathogenic fungi in fruits may be related to its ability to induce increased production of, e.g., polyphenol oxidase and peroxidase [42].

As research progressed, hypotheses on the mechanism of chitosan inhibition were put forward, but the exact mechanism of inhibition has still not been determined. Feng et al. [43] measured the OD of cell contents under UV light at a wavelength of 260 nm and showed an increase in OD. Using transmission electron microscopy to observe changes in the ultrastructure of *E. coli* and *S. aureus* before and after the action of chitosan, it was observed that the cell membrane was broken, accompanied by a large amount of content spillage (DNA and mRNA) at the periphery and bacterial apoptosis. Similarly, Hui et al. [44] evaluated the antibacterial activity of chitosan acetate solution against *E. coli* and *S. aureus*. The morphology of bacteria after chitosan treatment was observed by transmission electron microscopy. The integrity of the cell membranes of both bacteria and the permeability of the inner and outer membranes of *E. coli* were investigated by measuring the absorption values at 260 nm UV, fluorescence changes in cells treated with the fluorescent probe 1-N-phenylnaphthylamine, and the release of cytoplasmic h-galactosidase activity. The results showed that chitosan ultimately damaged the bacteria by increasing the permeability of the inner and outer cell membranes, releasing the cell contents. This damage is most likely caused by electrostatic interactions between the NH^3+^ group in the chitosan acetate solution and the phosphoryl group of the phospholipid fraction of the cell membrane. Imelda et al. [29] confirmed the disruptive effect of chitosan on protein synthesis by β-galactosidase expression assay.

The above studies showed that chitosan can inhibit bacteria by disrupting bacterial cell membranes and protein synthesis. Ming et al. [45] pointed out that chitosan has high chelating ability for various metal ions (including Ni^2+^, Zn^2+^, Co^2+^, Fe^2+^, Mg^2+^, and Cu^2+^) under acidic conditions. As a chelating agent, chitosan selectively chelates metal ions, thus playing a key role in microbial growth by inhibiting the growth and reproduction of microorganisms. Gooday et al. [46] pointed out that when chitosan crosses the cell wall of fungal pathogens with plant hydrolases as host, it penetrates the nucleus of the fungus, and the positively charged chitosan interacts with the negatively charged DNA, affecting RNA transcription and protein synthesis, thus achieving fungal inhibition.

In summary, there are three generally accepted theories about the mechanism of chitosan inhibition, the first of which is inter-charge interactions. As shown in Figure 3 [17], the electrostatic interaction between R-NH^3+^ of chitosan and negatively charged groups on the bacterial surface destabilizes the structure of the cell membrane, and the leakage of intracellular material leads to microbial death. The second theory concerns the chelating property of chitosan with metal ions. Chitosan chelates metal ions on the surface of bacteria, forming a microbially unavailable chelate, thus inhibiting microbial growth. The third theory holds that chitosan enters the cell and binds to DNA, and affects RNA transcription and protein expression, thus achieving the effect of bacterial inhibition. Although the above views are generally accepted, there are still many scholars who believe that the molecular weight and degree of deacetylation, and the differences between bacterial species, should also be considered when investigating the inhibition mechanism of chitosan.

### 3.2. Antioxidant Properties

The antioxidant properties of chitosan have been demonstrated [10,47,48]. It is well known that oxidation is one of the most important issues affecting the quality of meat, and heterocyclic amines, which are generated during the high-temperature processing of foods containing proteins, are carcinogenic substances. Many factors influence the formation of heterocyclic amines, mainly cooking methods, processing conditions (time, temperature), and the presence of antioxidant substances. Therefore, reducing heterocyclic amines by natural compounds and reducing or inhibiting the formation of these carcinogens in cooked meat by plant extracts containing antioxidants have become hot research topics [49].

When chitosan is added to meat products as a food additive, chitosan concentration and temperature are important factors that affect the experimental results. Fatih et al. [50] investigated the effect of adding different concentrations of chitosan (0.25, 0.50, 0.75, 1% *w*/*w*) to meatballs cooked at different temperatures on the formation of heterocyclic aromatic amines and the quality of the meatballs. The results showed that the heterocyclic amine content of the meatballs increased with increasing temperature (150, 200, and 250 °C), and the heterocyclic amine content showed a decreasing trend with increased chitosan concentration at the same temperature. Similarly, a study by Hojat et al. [51] showed that the addition of chitosan to huso fillets for cooking was effective in reducing heterocyclic amine production. Given that the production of heterocyclic amines usually requires high-temperature conditions, chitosan coatings or films have been less studied for reducing the production of heterocyclic amines. Kader et al. [52] studied four types of chitosan coating with different degrees of deacetylation and molecular weights for cherry preservation and evaluated the changes in total phenolic content, antioxidant capacity, total anthocyanin content, ascorbic acid, total pectin content, hardness, and color of cherries. The results showed that the antioxidant capacity of chitosan and the ascorbic acid content in cherries had a trend of increasing and then decreasing with the degree of deacetylation and molecular weight, and the chitosan with 81.22% deacetylation and 273 kDa molecular weight at 4 °C showed the best antioxidant capacity.

Relatively stable DPPH radicals have been widely used to test the ability of compounds to act as free radical scavengers or hydrogen donors [53]. The IC_50_ value is the concentration at which 50% free radical scavenging is obtained, and the lower the IC_50_, the more active the sample is as an antioxidant compound and the greater its ability to absorb free radicals. Rainy et al. [54] investigated the antioxidant properties of edible chitosan–galactose complex coating by compounding chitosan and galactose (0, 0.5, 1, and 1.5 g), and performed an in vitro test to evaluate the coating and analyze the parameters of antioxidant activity (DPPH method). The results showed a decreasing trend in the IC_50_ values, followed by an increasing trend. The lowest IC_50_ value of 43.20 ppm was recorded for the combination of chitosan and 1 g of galactose, which was the best antioxidant in the tested treatments. Similarly, Zhang et al. [55] reported the results of free radical scavenging experiments with different molecular weight chitosan, showing that high-molecular-weight chitosan had a high scavenging effect on hydroxyl radicals, and low-molecular-weight chitosan had a better scavenging effect on superoxide anion radicals and DPPH.

Giuseppina et al. [10] studied the effect of chitosan-based coatings on the freshness of figs by assessing the activities of enzymes, such as catalase, ascorbate peroxidase, polyphenol oxidase, and peroxidase. The results showed that the addition of chitosan coating significantly increased the total polyphenol, anthocyanin, and flavonoid contents and the antioxidant activity of stored figs, reduced oxidative stress, and prevented browning reactions compared with the untreated group. Miriam et al. [56] studied the effect of nano-chitosan/propolis coating on the shelf life and antioxidant capacity of strawberries, and the results showed higher total phenol and flavonoid content and antioxidant capacity of strawberries in the nano-chitosan/propolis coated group than in the untreated group at the end of the storage period. Chitosan delays the ripening and aging process of food, which extends the shelf life; largely maintains all sensory qualities of food; reduces enzymatic browning; decreases water loss; maintains the bright color, taste, and texture of food; and makes the aroma more durable, thus effectively maintaining the commercial value of food. There are many factors involved in the antioxidant performance of chitosan, and the main research concerns are concentration, degree of deacetylation, molecular weight, and antioxidant enzymes.

As the antioxidant mechanism of chitosan has been investigated, hypotheses have been proposed. Chitosan films in combination with other plant-based flavonoids have antioxidant properties as coatings on food surfaces [57]. Yuntao et al. [13] showed that high-molecular-weight chitosan films have a denser structure and better antibacterial properties. Braber et al. [58] concluded that chitosan biopolymers scavenge free radicals or chelate metal ions through a hydrogen or lone pair electron donor mechanism. The -OH and -NH_2_ groups in chitosan are the key functional groups for its antioxidant activity [59]. Zhang et al. [60] analyzed the barrier properties of chitosan–cyanidin films and concluded that the increased hydrogen bonding between cyanidin and chitosan molecules leads to a tighter arrangement between molecules inside the film, which improves its gas barrier properties.

The comprehensive analysis of the above studies led to three conjectures regarding chitosan’s antioxidant properties. The first is the barrier effect: chitosan coating can act as a barrier on the fruit surface, changing the internal gas atmosphere, reducing water loss, and delaying ripening. Second, many oxidation processes are carried out with the participation of metal ions, which play a role in transferring electrons during valence changes, and chitosan removes metal ions through chelation, thereby inhibiting oxidation reactions. Third, high-molecular-weight chitosan has denser intramolecular hydrogen bonds, which further prevents oxidation reactions from occurring when food comes into contact with air.

### 3.3. Enzyme Activity Inhibition

Enzymatic reactions play a non-negligible role in the softening and browning of foods. Tissue browning is inevitable in some fruits and vegetables with damaged surfaces due to the action of polyphenol oxidase. He et al. [18] studied the effect of clove oil–chitosan coating on the quality of fresh-cut lemons at four temperatures of 0, 4, 7, and 10 °C. By analyzing the changes in peroxidase, polyphenol oxidase, and phenolic acids, it was concluded that the chitosan coating had a stronger inhibitory effect on enzyme activity as the temperature decreased. Seafood is also highly susceptible to deterioration due to enzymatic reactions, which is the main reason for its quality decline [17].

Chitosan concentration is another important factor influencing enzyme activity. A-Dan et al. [61] investigated the effect of chitosan on papain and showed that with increased chitosan concentration, there was an inhibitory effect on papain when the concentration was greater than about 8.0288 g/L. Arisa et al. [62] investigated the effect of chitosan coating with different molecular weights at a concentration of 1% on the physicochemical properties of red bananas. The results showed that chitosan coating was able to reduce fruit respiration rate, ethylene production, and pectin hardness by inhibiting the activity of cell wall degrading enzymes (polygalacturonase and pectin lyase) that are important in pectin degradation, and with increasing molecular weight, chitosan coating showed better inhibition of enzymatic activity, thus delaying banana spoilage and deterioration. Although the mechanism by which chitosan inhibits the browning of fruits and vegetables and softening of meat tissues has not been elaborated, it must be related to the inhibition of enzymes associated with the occurrence of enzymatic reactions in foods.

With the study of the mechanism of chitosan’s enzymatic activity inhibition, related hypotheses have been proposed. Kurita et al. [63] indicated that polycations may compete with divalent metals, such as Mg^2+^ and Ca^2+^, present in the cell wall, thus disrupting the integrity of the cell wall or affecting the activity of degradative enzymes. To resist local browning in fruits and vegetables, chitosan binds suspended polyphenol oxidase molecules through electrostatic interactions to inactivate the enzyme, and the inactivated polyphenol oxidase is unable to transfer electrons directly to molecular oxygen, thus acting as an anti-browning agent [56]. Riaz et al. [64] studied the effect of chitosan-based apple peel polyphenol composite coating on improving the storage quality of strawberries. The antioxidant content of the coated fruits was relatively stable compared to uncoated fruits. The reduced antioxidant activity during storage may lead to decay and senescence. The ability of chitosan-based apple peel polyphenol composite coating to reduce decay, decrease enzymatic activity, and retain the quality attributes of fruits, leading to the degradation of antioxidant compounds, is closely related to the presence of efficient oxygen radical scavengers [65]. Yage et al. [66] used chitosan/nano-titanium dioxide coating to induce film formation on the surface of mangoes, creating a microenvironment that reduced decay and water loss and delayed respiratory peaks to preserve fruit flavor.

In summary, the hypotheses on the mechanism of action of chitosan’s enzyme activity inhibition include the following: First, the chelation of metal ions disrupts the enzyme activation pathway, and calpain and matrix metalloproteinases are Ca^2+^ dependent endogenous enzymes. Second, chitosan molecules penetrate the sarcoplasm and bind directly to the enzyme, thereby affecting enzyme activity. Third, chitosan may bind to structural proteins, which can affect the degradation and dissociation of enzyme proteins. Fourth, the chitosan coating alters the microenvironment of the sample, thus synthetically modulating the biochemical characteristics of the preserved food. Unfortunately, there are not enough relevant studies, and more research is needed to develop systematic interpretations of these hypotheses.

### 3.4. Biodegradability

Biodegradable polymer materials can be degraded by microorganisms or their secretions under the action of enzymes or chemical decomposition with a certain time and under certain conditions [67]. The degradation of biodegradable materials is generally accompanied by changes in their chemical and physical properties, reduced molecular weight, and the production of low-molecular-weight substances (CO_2_, H_2_O, CH_4_). Ruchir et al. [20] reported that chitosan coatings are environmentally friendly, biodegradable, and in most cases edible. Aris et al. [19] pointed out that although polymers synthesized with chitosan cannot be completely degraded, mixing chitosan with synthetic polymers can improve the decomposition rate of plastics that are more difficult to decompose. Polyvinyl alcohol (PVA), which is nontoxic and water-soluble, is one of the most commonly used synthetic polymers for chitosan. Yueming Li et al. [68] studied the effect of biodegradable, antibacterial chitosan starch composite film on the freshness of red grapes. The results showed that the biodegradable film had good water retention and antimicrobial efficacy and could be applied to grape preservation.

Zhang et al. [69] observed a link between degradation rate and molecular weight, degree of deacetylation, and distribution of N-acetyl D-glucosamine residues. Chitosan is a semi-crystalline polymer, and the relationship between chitosan biodegradability and the degree of deacetylation also depends on the degree of crystallinity, which reaches a maximum at a deacetylation degree of 0% (in the form of titin) or 100% (fully deacetylated chitosan) and decreases at intermediate values, with the rate of biodegradation increasing as the degree of crystallinity decreases. Similarly, Croisier et al. [70] found that acetyl residues distributed along the chitosan chain also affect the crystallinity of chitosan and, thus, the biodegradation rate. It can be concluded that smaller chitosan chains biodegrade more efficiently than higher-molecular-weight chitosan. In vivo, chitosan can be degraded by several enzymes, including lysozyme, a nonspecific enzyme present in all mammalian tissues. The degradation products are nontoxic oligosaccharides, and in vitro degradation of chitosan occurs by oxidative, chemical, or enzymatic hydrolysis. These methods are commonly used for low-molecular-weight chitosan prepared under controlled conditions.

In addition to being a polymer with an amino group, chitosan is also a polysaccharide and therefore contains readily breakable glycosidic bonds. Chitosan appears to be degraded in vivo by nonspecific enzymes, but mainly lysozyme has been reported to have this property. The biodegradation of chitosan leads to the formation of nontoxic oligosaccharides. These oligosaccharides, with variable lengths, can potentially be incorporated into metabolic pathways or excreted from the body [71,72]. Therefore, it is clear that chitosan is a strong natural candidate to replace nondegradable plastics in the future and its position as a natural alternative is established. Based on the current research, the relative application of chitosan in food packaging will be developed and commercialized shortly.

## 4. Status of Research on Chitosan Preservation

### 4.1. Enhanced Freshness Preservation Performance

Despite the many biological activities of chitosan described above, its antimicrobial and antioxidant capacities are still lacking when it is used for food preservation [73,74]. To address these issues and expand its application, many methods have been investigated to enhance its preservation performance, mainly including derivatization, chitosan nanoparticles, and combinations with natural extracts, essential oils, natural polymers, synthetic polymers, etc.

#### 4.1.1. Chitosan Derivatization

The functional groups of chitosan contain an amino group and two hydroxyl groups, which can easily react with a variety of chemical groups. Introducing new groups by modifying chitosan molecules to produce derivatives with excellent physical and chemical properties is important to expand the applications of chitosan and its derivatives [75]. Previous and current studies have shown that chitosan derivatives have positive effects in terms of antioxidant activity. Xiao et al. [76] studied the use of gallic acid–chitosan derivatives to preserve cherry tomatoes and evaluated the antioxidant and endogenous enzyme activities. The results showed that the enzyme-grafted gallic acid–chitosan derivatives had excellent antioxidant capacity in scavenging DPPH, hydroxyl, and superoxide anion radicals. This resulted in prolonged fruit ripening, reduced weight loss, high hardness, and little change in epidermal color in the treated group, and aromatic compounds remained relatively constant throughout storage due to delayed postharvest senescence.

Similarly, Neslihan et al. [77] synthesized two novel chitosan–Schiff base derivatives by condensation reactions of high- and low-molecular-weight chitosan with cotton phenol and studied their scavenging activity against 1,1-diphenyl-2-picrylhydrazyl (DPPH) radicals. They concluded that both chitosan–cotton phenol derivatives had better scavenging ability against DPPH radicals than unmodified chitosan. The modified chitosan had significantly improved antibacterial activity against foodborne bacterial spoilage. Similarly, chitosan derivatives have good film-forming and antimicrobial properties, and the use of film coating treatment has significant effects on inhibiting browning, reducing enzyme activity, maintaining fruit and vegetable quality, and prolonging freshness. Meriem et al. [78] prepared active nanocomposite films of cellulose nanocrystals reinforced with styrene-based quinoxaline-grafted chitosan by the solvent casting method and studied their antibacterial activity against five common bacteria. The films demonstrated good antibacterial activity against *Pseudomonas aeruginosa*.

Furthermore, films prepared from chitosan derivatives have different physical properties depending on the type and degree of substitution of the grafting groups. Bingnan et al. [79] found that the mechanical properties of chitosan products could be improved with a small amount of differently structured formyl saccharides by establishing intermolecular formyl–sugar bonds. When the amount of sucralfate and cottonseed aldehyde was 0.5%, the breaking strength and strain of chitosan films increased from 28 to 38 MPa and 19% to 48%, respectively, and the wet stability and toughness were improved at the same time. Currently, various chitosan derivatives have been successfully designed and applied to food preservation, as shown in Table 1. There are not many research cases of chitosan derivatives for food preservation due to the concern for their potential toxicity. Therefore, future research should not only explore more chitosan derivatives with specific and enhanced functions, but also develop green chemical processes to avoid toxic residues.

#### 4.1.2. Chitosan Nanoparticles

Due to its large particle size, chitosan does not come into contact with food spoilage factors when used for food preservation, thus limiting the biological activity of chitosan alone when used for this purpose [88]. Studying chitosan nanoparticles is another efficient option to enhance the utilization of chitosan. Melo et al. [89] developed composite pectin films supplemented with copper algae mud chitosan nanoparticles and evaluated their physical and mechanical properties. The mechanical analysis of maximum stress and elongation showed that nanoparticles as fillers increased the toughness of the pectin films. Water vapor permeability tests showed that nanostructured films containing copper sulfur had better barrier properties. More studies on using chitosan nanoparticles to enhance the physical and mechanical properties of chitosan films are presented in Table 2.

Ran et al. [90] studied the use of chitosan nanoparticles for fish preservation and found that the storage life of fish fillets wrapped in composite nanofilms could be extended by 6–8 days through sensory evaluation, microbiological analysis, pH, total volatile alkaline nitrogen value, thiobarbituric acid value, color, texture, and other storage quality indicators. The plant essential oil anthocyanin composite film had the best effect on fish fillet preservation, and the anthocyanin chitosan composite nanoparticle film had the best effect on protecting fish fillet appearance. In another study, modified magnetic chitosan nanomaterials obtained by combining modified chitosan with magnetic materials retained the properties of chitosan with the strong chelating ability of metal ions and had good regenerative properties, greatly expanding the application prospects of chitosan [91].

There are several methods to prepare chitosan nanoparticles, including the ionic gel [92], microemulsion [93], emulsification solvent diffusion [94], polyelectrolyte complex [95], and reversed-phase micellar [96] methods. Chitosan nanoparticles have both the biological properties of chitosan and the small size of nanoparticles with a large contact area. The final performance of nanoparticles also depends on several other parameters, such as the composition of secondary materials in the matrix, the material concentration/ratio, and the reaction time [97]. In addition, studies have reported using different types of chitosan (varying by molecular weight, degree of deacetylation, etc.) [98,99]. The applications of chitosan-based nanoparticles are gradually increasing, and it is expected that chitosan-based nanoparticle formulations will enter the market soon.

**Table 2 foods-11-01490-t002:** Correlation of mechanical properties of chitosan nanoparticles after film formation.

Property	Nanomaterials	Reference
Barrier	Chitosan/tripolyphosphate	[100]
Water sorption	Chitosan/starch	[101]
Stretchability	Chitosan/chlorogenic acid	[102]
Water vapor	Chitosan/TiO_2_ and Ag	[103]
Transparency	Chitosan/curcumin	[104]
Breathability	Chitosan/polyoxyethylene	[105]

#### 4.1.3. Chitosan Plant Extract Compound

Plant extracts are often rich in low-molecular-weight bioactive ingredients, such as polyphenols, terpenoids, and terpenoids, which are considered to be powerful antibacterial and antioxidant agents [106]. Plant extracts are known not only to act as free radical scavengers in vitro but also to protect the body from free radical activity. The respiration and transpiration of fruits and vegetables during storage can cause water loss, and when water loss reaches 5%, wilting and withering can occur, which can seriously affect the edible value [107]. Table 3 summarizes some recent literature on the use of plant extracts in combination with chitosan-based coatings/films for different food products.

The use of chitosan-based composite coating and preservation solution can reduce water loss, inhibit cellular action, and prolong the storage time of fruits and vegetables. Rambabu et al. [108] investigated the application of mango leaf extract incorporated into chitosan film to enhance the antioxidant activity and characterized the tensile strength. They found that the mango leaf extract–chitosan composite film had better tensile strength (maximum tensile strength of 23.06 ± 0.19 MPa) and less elongation compared to the pure chitosan film (18.14 ± 0.72 MPa). They also evaluated the antioxidant activity of chitosan films in terms of total phenolic content, DPPH radical scavenging ability, iron ion reducing ability, and ABTS radical scavenging ability, and the results showed that the antioxidant activity of increased with the addition of mango leaf extract. Yang et al. [109] conducted similar experiments, in which blueberry leaf extract was added to chitosan coating to maintain the postharvest quality of fresh blueberries. The composite film synthesized with chitosan and plant extracts also had UV-blocking and antioxidant abilities [110]. Wanli et al. [111] studied the process of preparing chitosan–banana peel extract composite film. Different contents of banana peel extract (4%, 8%, and 12%) were added to chitosan membranes using various characterization methods, and the experimental results showed that the chitosan–banana peel extract composite membranes had good antioxidant activity in different food samples.

Plant extracts, when synergized with chitosan, usually have an inhibitory effect on bacterial growth, which is related to metabolic disorders caused in bacteria through disruption of cell membranes, enzyme systems, or genetic material. Ali et al. [112] investigated chicken meat coated with chitosan and containing 1% essential oil of oregano and 1% or 2% grape seed extract stored in the refrigerator. On the 20th day of refrigeration, the minimum viable count of each treatment organism (Enterobacteriaceae, *Pseudomonas* spp., *Lactobacillus*, and *Saccharomyces* (yeast-mold)) was 3.54–4.51 log CFU/mL. Compared with the untreated group, the results indicated that the compound coating was effective in retarding microbial growth and oxidative activity. At present, most of the plant extracts compounded with chitosan are from easily available and nontoxic substances, such as herbs and spices, and they are used in food preservation by obtaining their essential oils, which have active antioxidant and antibacterial active properties when compounded with chitosan. However, there are still problems, such as the complex extraction process, low efficiency, and the effect on the sensory quality of food. Therefore, the extraction and bio-preservation ability of plant extracts still need to be explored in the future in order to provide better technical support for the development of food preservation.

**Table 3 foods-11-01490-t003:** Studies on chitosan–plant extract complexes for food coatings.

Commodity	Plant Extract	Reference
Blueberry	Blueberry leaf	[109]
Banana	Gum arabic	[113]
Sierra fish fillets	Tomato	[114]
Strawberry	Peony	[115]
Mango	Pomegranate peel	[116]
Apple	Banana peels	[117]
Sweet basil leaves	Thyme volatile oil	[118]
Le Conte pear	Beeswax–pollen grains	[119]

### 4.2. Freshness Preservation Technology

In recent years, the preparation of chitosan-based preservation films has become increasingly standardized, and the growing demand for food preservation has greatly contributed to the development of chitosan biofilm preservation technology. In terms of operation, traditional coating and film-making methods for food preservation have been widely used and have shown reliable results. The thickness of chitosan films is usually in the range of 6–80 µm, with an average of 25 µm, as reported by Elena et al. [120]. In addition, emerging technologies, such as film production from plant extracts, by lamination with plastic films, and by laminated layer self-assembly (LbL), are emerging. Both direct coating and bioactive packaging film preservation methods are systematically discussed at a technical level below.

#### 4.2.1. Coating Preservation

Coating preservation refers to covering the food surface with a layer of chitosan solution, which can usually be applied by dip coating, electrostatic spraying, or brushing. Due to its outstanding advantages of convenience and operability, it is often used in food preservation research. For example, Yong et al. [121,122] used electrostatic spraying of chitosan solution to preserve strawberries, which prolonged their shelf life and slowed down the degradation of quality. Rui et al. [123] investigated the potential application of chitosan as a natural growth regulator for bean sprouts by dipping bean sprouts in different concentrations of chitosan solution. The results showed a significant increase in the hypocotyl length and fresh weight of bean sprouts compared to the blank control. In addition, chitosan solution combined with plant extracts have been used as dip coatings to retard the quality deterioration of fruits, vegetables, meat, etc. Mehran et al. [124] applied chitosan solution mixed with Berberis extract and peppermint essential oil to turkey breast meat, and microbial counts and oxidation levels were significantly reduced under refrigerated conditions.

Usually, chitosan-based dip-coating treatment consists of the following steps: First, a pre-determined concentration of chitosan solution is prepared, and then the food is immersed in the solution for a pre-determined time, depending on the characteristics of the food. Finally, the food is removed, the excess solution adhering to the surface is drained, and the product is dried in a specific airflow environment. Regarding other coating methods, all steps are similar to dip coating except for the intermediate steps. Sometimes, repeated coating is applied to enhance preservation during storage [125].

Compared to other coating methods, it is obvious that dip coating is easy to operate under simple conditions. However, it is also prone to problems with dilution and contamination in practical applications. In addition, gravity tends to cause uneven film thickness on the food surface and reduces productivity to some extent [126]. The preservation substrate determines the thickness of the coating, the equipment requirements, and the drying technology used. In addition, self-healing coatings are intelligent materials that can repair coating damage and restore its properties, reducing or eliminating the adverse effects of damage. As a method of fabricating multilayer coatings, laminated layer self-assembly (LbL) is often used to construct self-healing coatings. Yu et al. [127] investigated a self-healing coating by assembling chitosan (CS) with sodium alginate (SA) layer-by-layer, and the damaged area could be completely repaired after three assembly cycles. The self-healing schematic is shown in Figure 4. The mechanical properties, water barrier, and oxygen barrier of the repaired coating were 97%, 95%, and 63% of those of the intact coating, respectively, which basically restored the barrier properties of the intact coating, although the permeability increased. In addition, the coating reduced the effect of coating damage by restoring the barrier properties of the coating and extending the shelf life of the strawberries. Thus, it can be seen that chitosan coating not only slows down food quality degradation, but also demonstrates efficient antioxidant and antibacterial effects. However, the uniformity of covering food and rough sensory effects need to be further improved. Therefore, the application of chitosan coating for freshness is worth further study, as it has the potential to become a low-cost alternative technology for fresh food preservation.

#### 4.2.2. Film-Making and Preservation

Bioactive chitosan packaging films are made by co-blending chitosan solution with bioactive materials to make cling film in which to wrap food and provide freshness. Cling films can be produced in advance in large quantities without the condition of preserved products. They can be broadly divided into two types according to the composite materials: packaging film based on chitosan solution and hybrid film prepared in combination with food-grade plastic film.

Chitosan-based biofilms are often prepared by co-blending with extracts derived from natural plants, which are of increasing interest to the food industry because of their eco-safety and rich nutritional value. The addition of active ingredients not only enhanced the inherent bioactivity of the films, but also improved the final mechanical properties and applicability of the films. For example, Zhang et al. [128] improved the mechanical properties of chitosan films by vanillin modification and showed that the film stretch of vanillin/chitosan composite films increased by 53.3% and moisture permeability decreased by 36.5% compared to pure chitosan films. Ting et al. [129] added spice extracts to chitosan films to extend the shelf life of frozen pork. The data showed that chitosan could interact intermolecularly with the polyphenols in the spices and also caused the formation of covalent bonds between the hydrogen and water molecules of the polyphenols, increasing the hydrophilicity of the composite films. In addition, plant essential oil, an aromatic oily liquid extracted from plant tissues and organs, can not only scavenge free radicals in the body and perform antioxidant functions, but also has the functions of regulating the balance of intestinal flora, killing pathogenic bacteria, and promoting the secretion of digestive juices, which is a plant extract with more applications at present. When the plant essential oil is compounded with a chitosan solution in a certain ratio, the prepared chitosan film will also have the relevant nutritional properties of plant extracts, and the plant extracts can function at low concentrations. The selective gas (CO_2_ and O_2_) permeability and good mechanical properties of chitosan make it an excellent film-forming material. However, the low water barrier properties of chitosan films limit their application, as effective control of water transfer is a desirable property for most food packaging, especially in humid environments [130]. Several studies have focused on the moisture permeability of chitosan films containing essential oils or other plant extracts. López et al. [131] found that the addition of carvacrol (0.5, 1.0, and 1.5% *v*/*v*) significantly reduced the moisture permeability of chitosan films. When essential oils or other plant extracts (tea tree, carvacrol, cinnamon, turmeric, etc.) were added to chitosan films, the moisture permeability was significantly reduced, which may be due to the hydrophobicity of the essential oil particles. It can be seen that various properties of the film or coating can be modified by adding essential oils, and chitosan with essential oils has been shown to increase hydrophobicity, reduce water vapor permeability, and improve the antimicrobial activity of the film.

For the preparation of multilayer or hybrid structured packaging films in combination with food-grade plastic films, synthetic plastics and biodegradable polymers provide sufficient mechanical strength to meet the packaging requirements and are widely used matrix materials [132]. With regard to structured packaging films, the physical and mechanical properties depend heavily on the mixing state and compatibility between the constituent components. Shun et al. [133] prepared chitosan–polylactic acid plastic films by an extrusion method and used them to preserve grouper fillets, then analyzed their physical properties and antibacterial activity. The results showed that the water vapor transmission rate and water content of the plastic films was increased, and inhibition of *Escherichia coli*, *Pseudomonas fluorescens*, and *Staphylococcus aureus* reached over 95%. For hybrid structured packaging films, the thermal stability of the active component is a key factor to consider. Sudharsan et al. [133] prepared a compatible PLA/chitosan composite film with a thickness of 0.25 mm by the solvent casting method, and it demonstrated good thermal stability, low oxygen permeability, and high mechanical properties. Although all the above studies are still at the experimental stage and still some distance from commercialization, it is believed that there will be a positive development trend of bioactive chitosan packaging films through the continuous development of film making technology.

## 5. Conclusions and Outlook

In summary, chitosan has good biological activities, including antibacterial and antioxidant activity and enzyme activity inhibition, and has demonstrated great potential in the field of food preservation. Thus, it has attracted widespread attention. Although there are inconsistent views among domestic and foreign scholars on the mechanism behind chitosan’s preservation of food, the generally accepted views are as follows: (1) inter-charge interaction, (2) chelation of metal ions, (3) disruption of cell membranes to enter bacteria, bind DNA, and affect protein expression, (4) dense intramolecular hydrogen bonding of chitosan, and (5) binding to enzymes or structural proteins, which affects enzyme activity. In addition, chitosan preservation methods are constantly changing and will continue to be integrated.

This review is a discussion and summary of the current status of chitosan in the field of food preservation, to help the reader understand its importance as a sustainable material in this field and related strategies to improve its properties to enlighten future developments. While much work has been done to establish methods and elucidate the positive performance of chitosan in food preservation, there are still many difficulties and theoretical shortcomings in practical application. Therefore, future developments should focus on the following aspects: in-depth research on the mechanism behind chitosan’s food preservation properties, and technical optimization and enrichment of strategies to further approach practical applications. In addition, studies analyzing the synergistic effects between components and methods as well as potential toxicity are also needed. Finally, the advantages of various technologies and composite components should be summarized regularly to help achieve effective the targeted inhibition of food spoilage sources. The future of chitosan as a sustainable material in the food preservation industry is bright, and expanding its application will be an inevitable trend as the technology continues to be developed.

## Figures and Tables

**Figure 1 foods-11-01490-f001:**
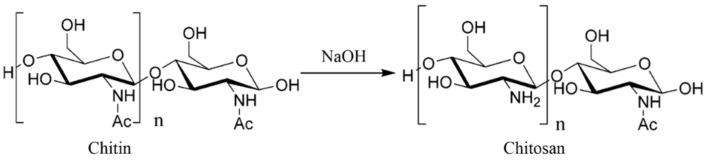
Preparation of chitosan from chitin by deacetylation.

**Figure 2 foods-11-01490-f002:**
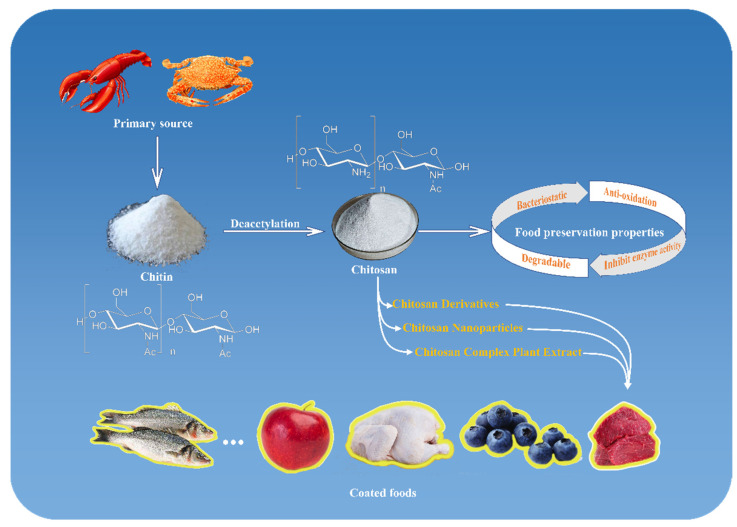
Sources, properties, and applications of chitosan.

**Figure 3 foods-11-01490-f003:**
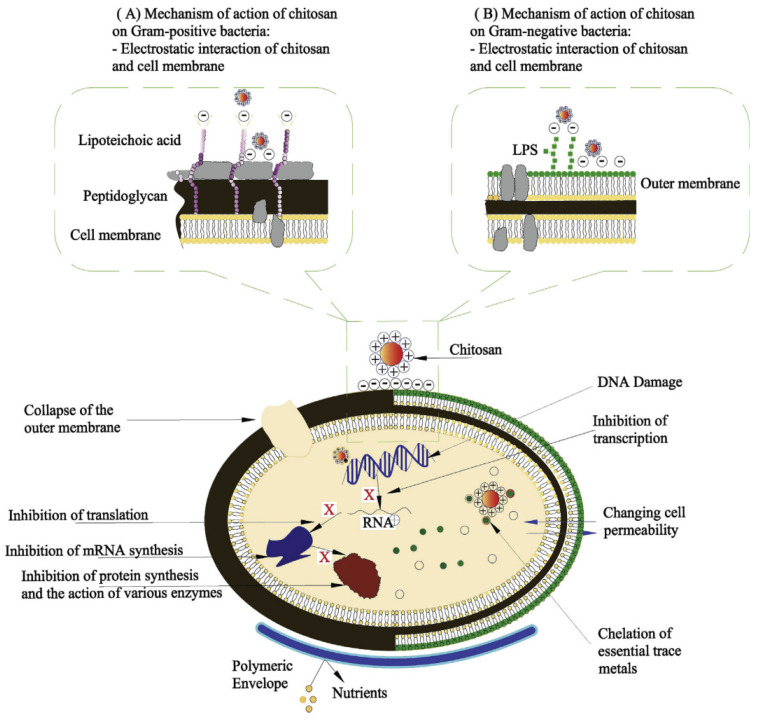
Mechanisms used to explain antibacterial activity of chitosan.

**Figure 4 foods-11-01490-f004:**
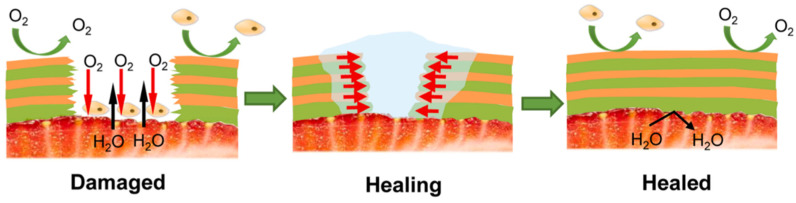
Diagram of self-healing behavior of chitosan (CS)/sodium alginate (SA) coating.

**Table 1 foods-11-01490-t001:** Related studies on chitosan derivatives for food preservation.

Product	Derivative	Reference
Beef	Chitosan/lauric	[80]
Tomato	N, O-carboxymethyl chitosan	[81]
Cucumber	Chitosan-g-salicylic	[82]
Strawberry	N-succinyl chitosan	[83]
Mulberry	Chitosan-g-caffeic	[84]
Blueberry	Carboxymethyl chitosan-peptide	[85]
Button mushroom	Gallic acid-grafted chitosan	[86]
Pork	Ferulic acid-grafted chitosan	[87]

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
