# Peer review of "Chitosan-Based Materials: An Overview of Potential Applications in Food Packaging"

_foods, 2022, doi:10.3390/foods11101490_

Round 1

Reviewer 1 Report

The authors present a collection of single results whose interrelationship is not worked out consistently and which are presented separately in a very imprecise and superficial manner. There are attempts to present specific mechanisms, but an overarching logical concept is missing.  Also, the difference between pure research results in the laboratory and a possible use in practical applications is not addressed. It is also not particularly innovative to present the methodology of a literature search, which should be well known to all scientists (lines 57 - 60).

Starting with the title: It is misleading, because active packaging for food is not properly presented in the article at all, but only together with other methods of preservation such as direct coatings of products. The differences between these approaches are not clearly presented and in many cases the reader does not even recognize whether it is just a coating, an impregnation or a real packaging made of film materials (e.g. lines 64 - 66, line 131).

Section 3.3 contains long-known generalities, e.g., the relationship between low crystallinity and high biodegradability.

Sections 4.2.1 and 4.2.2 are even more extreme in their unstructured accumulation of a wide variety of facts and also repeat numerous points from previous sections.

The sheer style is also anything but professional:

Chapter 2 begins with a terribly worded monster sentence (lines 51 - 57)

There are many repetitions, some in short (lines 23 - 25, 38 - 40, 129) intervals, some in longer ones (e.g. lines 105 - 107 vs. lines 204 - 207 and lines 260 - 261 vs. lines 267 - 270).

Many citations are written as fragments and one does not know exactly to which sentence to assign them (lines 137, 160, 166, 187, 219, 418). A heading ("Patents") also appears without finding any associated text (line 527).

Some sentences are tautologies or represent self-evident facts (lines 235 - 236, 311 - 312).

Lines 199 - 202 make all previously made statements relative.

In addition, there are some glaring errors that cast doubt on the authors' expertise:

Chitin, not chitosan, is found in shrimp and crabs (line 22).

Chitosan is not widely used in food. In many countries, it is not even approved as a food additive (line 28).

Light is absorbed, not the optical density (line 87).

Since when does chitosan contain ascorbic acid? (line 134)

Catalase, peroxidase, etc. are anything but antioxidant enzymes (line 143)

The manuscript, in order to become scientifically relevant, not only needs to be revised linguistically and editorially, but the connections need to be worked out much more clearly and concisely. Otherwise, it will be of very limited value to a scientist in the relevant field.

Author Response

Reviewer 1

  1. The authors present a collection of single results whose interrelationship is not worked out consistently and which are presented separately in a very imprecise and superficial manner. There are attempts to present specific mechanisms, but an overarching logical concept is missing. 

Response: Thank you very much for these valuable and constructive comments. Based on your comments, we have revised the article in depth, for the single series of results we have made a more standardized summary to make the article more in-depth, and for the proposed mechanism of chitosan bioactivity we have revised it in a more comprehensive and logical structure, and our changes are marked in red font in the revised manuscript for your reference. We hope that our revised manuscript will satisfy you.

  1. Also, the difference between pure research results in the laboratory and possible use in practical applications is not addressed.

Response: Thank you for your valuable comments. There is indeed a difference between pure research results in the laboratory and actual commercial applications, but after our extensive review of the literature, we found that there is indeed less literature on finished chitosan for commercial use, so the difference between the two is not perfectly addressed in this review, but the issue you mentioned is indeed a point worth thinking about.

  1. It is also not particularly innovative to present the methodology of a literature search, which should be well known to all scientists (lines 57 - 60).

Response: Thank you very much for your suggestion, the search method in this paper is indeed from the more general literature database, so we added the relevant industry more recognized search engine to the current search database to find the article, we appreciate your question for this aspect.

  1. Starting with the title: It is misleading, because active packaging for food is not properly presented in the article at all, but only together with other methods of preservation such as direct coatings of products. The differences between these approaches are not clearly presented and in many cases the reader does not even recognize whether it is just a coating, an impregnation, or a real packaging made of film materials (e.g. lines 64 - 66, line 131).

Response: Thank you very much for your suggestion. Since we did not include a separate section on active packaging in the article, we decided to change the title of the article to "Chitosan-based materials: an overview of potential applications in food packaging", and the changes are in lines 2-3 and 18, which are marked in red.

  1. Section 3.3 contains long-known generalities, e.g., the relationship between low crystallinity and high biodegradability.

Response: Thank you for your suggestion, we have checked the issues you raised in section 3.3 and we have removed and modified such general issues and conclusions, the modified parts are located in lines 261-263 and have been marked in red.

  1. Sections 4.2.1 and 4.2.2 are even more extreme in their unstructured accumulation of a wide variety of facts and also repeat numerous points from previous sections.

Response: Thank you very much for these valuable and constructive comments. For section 4.2.1 we have grouped a large number of examples. First: the outstanding advantages of the convenience and workability of the coating. Second: The combination of polysaccharide solution with plant extracts used as an impregnating coating is presented. Third: The steps related to the preparation of chitosan coatings are summarized. Fourth: The problems associated with chitosan coating are analyzed. For section 4.2.2 we focus on packaging films made based on chitosan solution and hybrid films prepared in combination with food-grade plastic films, where the advantages and disadvantages of both are presented and analyzed. The modified sections are located in lines 497-608 and have been marked in red.

  1. Chapter 2 begins with a terribly worded monster sentence (lines 51 - 57)

Response: Thanks to your suggestions, we have rewritten this section, and the changes are located in lines 63-64 and have been marked in red.

  1. There are many repetitions, some in short (lines 23 - 25, 38 - 40, 129) intervals, some in longer ones (e.g. lines 105 - 107 vs. lines 204 - 207 and lines 260 - 261 vs. lines 267 - 270).

Response: Thank you very much for your suggestion, we checked the above problem you mentioned, and we have removed and modified this repetitive problem by modifying some short intervals (lines 23-25 of the original text correspond to lines 23-24 now, lines 38-40 of the original text correspond to lines 39-40 now, and lines 129 of the original text correspond to lines 181-183 now) and long intervals (for example, lines 105-107 and lines 204-207 of the original text correspond to lines 282-284 now, which have been marked in red.

  1. Many citations are written as fragments and one does not know exactly to which sentence to assign them (lines 137, 160, 166, 187, 219, 418).

Response: According to your comments, we checked (lines 137, 160, 166, 187, 219, and 418) and found that there were indeed problems with large quotations, so we refined and rewrote these large quotations, and the modified parts are located in (lines 202-206、227-238、241-244、261-263、295-300、505-508), which have been marked in red.

  1. A heading ("Patents") also appears without finding any associated text (line 527).

Response: Thank you very much for your suggestion. We checked line 527 and found that it was indeed a redundant title, which has been removed in our revised version and marked in red in line 638.

  1. Some sentences are tautologies or represent self-evident facts (lines 235 - 236, 311 - 312).

Response: We have checked lines 235-236 and 311-312 and found that there are indeed repetitive sentences with the same meaning, which have been deleted in our revised draft, and the revised parts are located in lines 338-340 and 389-391, which have been marked in red.

  1. Lines 199 - 202 make all previously made statements relative.

Response: Based on your suggestion we looked at lines 199-202 and we explained in lines 279-300 that the potential reason for chitosan inhibition of enzyme activity could be that chitosan inhibits enzyme activity associated with the browning of fruits and vegetables.

  1. Chitin, not chitosan, is found in shrimp and crabs (line 22).

Response: Based on your suggestion, we have checked the sentence in line 22 and have made changes in our revised draft, which is located in line 22 and has been marked in red.

  1. Chitosan is not widely used in food. In many countries, it is not even approved as a food additive (line 28).

Response: Thank you very much for your suggestion, we have reviewed the relevant information and found that chitosan in food only has a large number of studies rather than applications, we have made changes in the revised draft, the revised part is located in lines 28-30, has been marked in red.

  1. Light is absorbed, not the optical density (line 87).

Response: Thank you very much for your valuable comments. We checked the sentence in line 87 and found that it was caused by an error in the presentation, which we have revised in the revised draft, and the revised part is located in lines 115-123 and has been marked in red.

  1. Since when does chitosan contain ascorbic acid? (line 134)

Response: Based on your comments we checked line 134 of the article and found that it was due to our presentation problem, and have now marked our changes in red in the revised draft, which are located in lines 181-183.

  1. Catalase, peroxidase, etc. are anything but antioxidant enzymes (line 143)

Response: Thank you very much for your comments. We have checked line 143 of the article and checked again what appears in the corresponding article and have now marked our changes in red in the revised version, which is located in lines 206-209.

  1. The manuscript, in order to become scientifically relevant, not only needs to be revised linguistically and editorially, but the connections need to be worked out much more clearly and concisely. Otherwise, it will be of very limited value to a scientist in the relevant field.

Response: We appreciate your suggestions, and we have made a number of changes to the manuscript to make it scientifically relevant, not only linguistically and logically, but also to analyze and summarize the mechanism, and we have enlisted the help of a native English-speaking colleague to correct the language and text. We hope that you will be satisfied with our revised version.

Reviewer 2 Report

Foods

foods-1716627

Chitosan-Based Materials: An Overview on The Potential Applications in Food Active Packaging

Dear Editor,

This paper reviews the research on the bioactive mechanism possessed by chitosan in recent years and introduces the application strategies of chitosan in the preservation and the modification of chitosan and different preservation techniques to explore the potential application value of chitosan-based food active packaging. The article has been well designed and written. Discussions are perfect. It can be accepted after necessary corrections done. My comments and questions;

-   In the literature, it has been shown that chitosan has inhibitory effect of the formation of some food toxicants such as heterocyclic aromatic amines due to its antioxidant effect. This can be mentioned in the text for better highlighting the antioxidant effect of chitosan.

-   Lines 187-189: Check the sentence!

-   Line 281: Give references for this sentence!

-   Please give some information about the IC50 value of chitosan for better understanding of the antioxidant effect.

-   For which foodstuffs are these film coatings more suitable?

-   Could you give information about the film thickness?

Author Response

Reviewer 2

This paper reviews the research on the bioactive mechanism possessed by chitosan in recent years and introduces the application strategies of chitosan in the preservation and the modification of chitosan and different preservation techniques to explore the potential application value of chitosan-based food active packaging. The article has been well designed and written. Discussions are perfect. It can be accepted after necessary corrections are done. My comments and questions;

Response: We greatly appreciate your encouraging comments on our manuscript. We have revised the manuscript based on these comments and suggestions, and have indicated our changes in red in the revised manuscript for your reference. We hope that our revised manuscript will be to your satisfaction.

  1. In the literature, it has been shown that chitosan has inhibitory effect of the formation of some food toxicants such as heterocyclic aromatic amines due to its antioxidant effect. This can be mentioned in the text for better highlighting the antioxidant effect of chitosan.

Response: Thank you very much for your professional suggestion, so to highlight the antioxidant effect of chitosan to inhibit the formation of heterocyclic amines in meat, we have added the relevant content in our revised manuscript and added two new related papers [49][50], the revised part is located in lines 162-178 and has been marked in red.

  1. Nadeem, H.R.; Akhtar, S.; Ismail, T.; Sestili, P.; Lorenzo, J.M.; Ranjha, M.M.A.N.; Jooste, L.; Hano, C.; Aadil, R.M.J.F. Heterocyclic aromatic amines in meat: formation, isolation, risk assessment, and inhibitory effect of plant extracts. 2021, 10, 1466.
  2. Oz, F.; Zaman, A.; Kaya, M.J.J.o.F.P.; Preservation. Effect of chitosan on the formation of heterocyclic aromatic amines and some quality properties of meatball. 2017, 41, e13065.
  3. Lines 187-189: Check the sentence!

Response: Based on your suggestion, we have checked the sentences in lines 187-189 and found that there are indeed improprieties, which have been revised in our revised draft, and the revised parts are located in lines 261-263, which have been marked in red.

  1. Line 281: Give references for this sentence!

Response: Based on your suggestion, we have checked the content at line 281, which provides a reference to the limited antimicrobial and antioxidant capacity of chitosan when used for food preservation and the application of chitosan films in food biodegradable packaging materials that are still deficient due to their low mechanical properties, and the modified part is located at lines 354-356 and has been marked in red.

  1. Liu, J.; Liu, S.; Wu, Q.; Gu, Y.; Kan, J.; Jin, C.J.F.H. Effect of protocatechuic acid incorporation on the physical, mechanical, structural, and antioxidant properties of chitosan film. 2017, 73, 90-100.
  2. Narasagoudr, S.S.; Hegde, V.G.; Vanjeri, V.N.; Chougale, R.B.; Masti, S.P.J.C.P. Ethyl vanillin incorporated chitosan/poly (vinyl alcohol) active films for food packaging applications. 2020, 236, 116049.
  3. Please give some information about the IC50 value of chitosan for better understanding of the antioxidant effect.

Response: Thank you very much for your professional opinion, so we looked for information about the IC50 value of chitosan, the lower the IC50, the more active the sample is as an antioxidant compound and the better the sample absorbs free radicals (50% of DPPH compounds). To better understand the antioxidant effect of chitosan, we have added content related to this in the revised manuscript, and the changes are located in lines 199-209 and have been marked in red.

  1. For which foodstuffs are these film coatings more suitable?

Response: Based on your questions we discussed that chitosan films are more suitable for preserving perishable berry fruits and vegetables as well as meat products that need to be preserved, where the list of chitosan derivative preservation products is shown in Table 1, at line 402 in the text, which is marked in red in the text.

  1. Could you give information about the film thickness?

Response: Following your proposal, we have provided the relevant thickness of the film, giving the range of application of the film, the additions are located in lines 489-450 and the modifications are marked in red.

Reviewer 3 Report

After reviewing the paper entitled "Chitosan-Based Materials: An Overview on The Potential Applications in Food Active Packaging", I need to required a major revision, primarily because the superficially processed subsection about antimicrobial activity.

Namely, the antibacterial activity are described in subsection 3.1., but non of name of bacteria are not italic, some crucial part of this subsection is inaccurate and reasonable presented. Please, rewrite the whole part. Bacteriostatic and bactericide effect is not the same.

Also, I need to required the subsection about antimicrobial activity against fungi and yeasts. This area is better and more widely presented in the scientific literature.

Author Response

Reviewer 3

  1. After reviewing the paper entitled "Chitosan-Based Materials: An Overview on The Potential Applications in Food Active Packaging", I need to required a major revision, primarily because the superficially processed subsection about antimicrobial activity.

Response: Thank you very much for these valuable and constructive comments. We have reorganized and revised this subsection in the revised draft, and the revised part is located in lines 61-109 and has been marked in red.

  1. Namely, the antibacterial activity are described in subsection 3.1., but non of name of bacteria are not italic, some crucial part of this subsection is inaccurate and reasonable presented. Please, rewrite the whole part. Bacteriostatic and bactericide effect is not the same.

Response: Thank you very much for your professional opinion, we have revised and rewritten the inaccurate and unreasonable part of the chapter on bacterial inhibition, paid attention to the italic writing norms of bacteria, and standardized the chapter on bacterial inhibition, the revised part is located in lines 67-125 and has been marked in red.

  1. Also, I need to required the subsection about antimicrobial activity against fungi and yeasts. This area is better and more widely presented in the scientific literature.

Response: Thank you very much for your professional opinion. We have reviewed a lot of literature and found that yeast and fungi are the most sensitive groups to chitosan, which is conducive to a better understanding of the antibacterial properties of chitosan. A section on the antifungal yeast of chitosan has been added to the revised manuscript, the modified part is located in lines 97-112, and new related literature [35-42] has been added and marked in red.

  1. Aider, M.J.L.-f.s.; technology. Chitosan application for active bio-based films production and potential in the food industry. 2010, 43, 837-842.
  2. Arceo-Martinez, M.T.; Jiménez-Mejía, R.; Salgado-Garciglia, R.; Santoyo, G.; Lopez-Meza, J.E.; Loeza-Lara, P.D.J.A. In vitro and in vivo anti-fungal effect of chitosan on post-harvest strawberry pathogens. 2019, 53, 1297-1311.
  3. Jackson, S.; Heath, I.J.M.R. Roles of calcium ions in hyphal tip growth. 1993, 57, 367-382.
  4. Kanawi, M.A.; AL Haydar, M.; Radhi, W.N.J.E.J.o.B. Effect of Chitin and Chitosan in Improvement of Plant Growth and Anti-Fungal Activity. 2021, 61, 513-519.
  5. Qin, Y.; Li, P.; Guo, Z.J.C.P. Cationic chitosan derivatives as potential antifungals: A review of structural optimization and applications. 2020, 236, 116002.
  6. Palma‐Guerrero, J.; Jansson, H.B.; Salinas, J.; Lopez‐Llorca, L.J.J.o.a.M. Effect of chitosan on hyphal growth and spore germination of plant pathogenic and biocontrol fungi. 2008, 104, 541-553.
  7. El Ghaouth, A.; Arul, J.; Grenier, J.; Asselin, A.J.P. Antifungal activity of chitosan on two postharvest pathogens of strawberry fruits. 1992, 82, 398-402.
  8. Pastor, C.; Sánchez-González, L.; Marcilla, A.; Chiralt, A.; Cháfer, M.; González-Martínez, C.J.P.B.; Technology. Quality and safety of table grapes coated with hydroxypropylmethylcellulose edible coatings containing propolis extract. 2011, 60, 64-70.

Round 2

Reviewer 1 Report

General remarks:

Looking at the number of errors still present and new ones added, I cannot believe that an expert native speaker proofread the manuscript.

If you want to quantify the effect of chitosan in coatings, always talk about the chitosan concentration in the coating solution. However, this is irrelevant as soon as a continuous film has been formed on the surface and only the thickness of the resulting film is then relevant. You still do not address this point.

The effect of chitosan as a coating or film should be carefully separated from its effect as a food additive (lines 152 to 163).

If you characterize chitosan coatings as "edible", you should check and comment on whether the authors of the respective original papers really tested the edibility (e.g. by sensory tests) or whether they only claimed it without proving it (which is unfortunately very common in the respective literature).

Now for the points of my previous report:

#1: Unfortunately, I cannot conclude that the corrections have substantially improved the manuscript

#2: Concerning my remark, you are writing: “…but the issue you mentioned is indeed a point worth thinking about” without any indication of appropriate changes. You should instead simply put the fact in a prominent place that the research on chitosan so far did practically not lead to commercially usable packaging products.

#3: Instead of simplifying section 2, you add another monster sentence (lines 49 to 56). Please shorten this section substantially and make it readable.

#4: I wrote: “The differences between these approaches are not clearly presented and in many cases the reader does not even recognize whether it is just a coating, an impregnation, or a real packaging made of film materials (e.g. lines 64 - 66, line 131).” You did not comment this remark and I do not see where you changed your text accordingly.

#5: The authors have not only revised 3.3. (slightly), but also apparently very extensively 3.1 and 3.2. (apparently mainly by copy and paste), but without saying what exactly they have done there. I find it very tedious for a reviewer to have to do his own text comparison here. In addition, new errors have been added: Lines 64 / 65 are again unsatisfactory: “… can effectively and slow down…” Verb is missing. “…the shelf life of food preservation“: No, the shelf life of food! In addition, one finds an incomplete citation in line 72, incomplete sentences in line 75, tautology in line 146.

#6: Sections 4.2.1 and 4.2.2 have been revised, but unfortunately have not been sufficiently improved as a result. In addition, some further errors have been added: Lines 452 to 455 read as if chitosan is already frequently found in practical applications. Unfortunately, this is not true. The term "LbL self-assembly" is written out as "layer-by-layer self-assembly" and not "laminated layer self-assembly". Lines 485 to 487 are contradictory: when a coating is damaged and repaired, its permeability to various substances increases and it does not decrease. Lines 507 / 508: What do nutritional values of plant extracts have to do with the properties of films? By the way, plant extracts can very well affect the sensory properties of foods, such as thyme or hops.

#7: The changes in lines 63-64 do not concern chapter 2, as chapter 3 already begins at line 59.

#8: The changes marked in red do not match the indicated line numbering or the changes are hidden in larger text blocks that have also been marked in red.

#9: Also here, the changes marked in red do not match the indicated line numbering

#11: Again, the changes marked in red do not match the indicated line numbering

#12: Your response does not match your new text

#14: Again, the changes marked in red do not match the indicated line numbering

#15: Again, the changes marked in red do not match the indicated line numbering

#16: The text in lines 181 – 183 has no relation to my question. Moreover, the sentence “Similarly, …” is incomplete

#17: Your answer to my remark cannot be found in the region of lines 206-209. Instead, the same false statement is now found in lines 185 – 187.

New remark: Check the sentence in lines 314 – 315 for correctness. (“…its limited … capacities are still lacking…”? Very peculiar meaning.)

Reviewer 3 Report

The authors has improved their work, all suggestions was involved in the revised version, so I can implicate the acceptance of the paper.

Author Response

Thank you very much for your valuable comments on the article.